
The seismogenic fault system of the 2017 $M_w$ 7.3 Iran-Iraq earthquake: constraints from surface and subsurface data, cross-section balancing and restoration

Stefano Tavani[1], Mariano Parente[1], Francesco Puzone[1], Amerigo Corradetti[1], Gholamreza Gharabeigli[2], Mehdi Valinejad[2], Davoud Morsalnejad[2], Stefano Mazzoli[1]

1 DISTAR. Università degli Studi di Napoli "Federico II". Napoli, Italy.
N.I.O.C., Tehran, Iran.
Corresponding author: Stefano Tavani, stefano.tavani@unina.it

**Abstract**

The 2017 $M_w$ 7.3 Iran-Iraq earthquake occurred in a region where the pattern of major plate convergence is well constrained, but limited information is available on the seismogenic structures. Geological observations, interpretation of seismic reflection profiles, and well data are used in this paper to build a regional balanced cross-section that provides a comprehensive picture of the geometry and dimensional parameters of active faults in the hypocentral area. Our results indicate: (i) coexistence of thin- and thick-skinned thrusting, (ii) reactivation of inherited structures, and (iii) occurrence of weak units promoting heterogeneous deformation within the Paleo-Cenozoic sedimentary cover and partial decoupling from the underlying basement. According to our study, the main shock of the November 2017 seismic sequence is located within the basement, along the low-angle Mountain Front Fault. Aftershocks unzipped the up-dip portion of the same fault. This merges with a detachment level located at the base of the Paleozoic succession, to form a crustal-scale fault-bend anticline. Size and geometry of the Mountain Front Fault are





consistent with a down-dip rupture width of 30 km, which is required for an $M_w$ 7.3 earthquake.

## Introduction

On November 12, 2017, a $M_w$ 7.3 earthquake struck the north-western portion of the Lurestan region of the Zagros Belt, at the boundary region between Iran and Iraq (Fig. 1). This earthquake had a thrust fault plane solution with a 351°-striking and 16°-dipping nodal plane. The other nodal plane has a strike of 122° and a dip of 79°. The P axis plunges 33° toward 223°, whereas the T axis plunges 54° toward 18° (Fig. 1) (Source: USGS,

https://earthquake.usgs.gov). These parameters indicate SW-directed co-seismic slip along a low-angle thrust, such a direction being nearly perpendicular to the strike of the Zagros Belt and of its main thrust systems. The hypocenter is located at a depth of ca. 20-km where, according to preliminary teleseismic data, the slip was nearly 9 m (Utkucu, 2017). Coherently with SW-directed motion along a gently dipping thrust, interferometric SAR data show a

NW-SE to NNW-SSE-elongated displacement field (Fig. 2). Consistently, the maximum surface deformation (reaching ca. 90 cm of uplift; Kobayashi et al., 2018) is shifted some tens of km SW-ward of the epicentre of the main shock. Forty-five $M_w > 4$ aftershocks followed during the next 30 days in a N-S-elongated, 50x150 km area located to the west of the main shock (Fig. 2). Aftershocks lined up, as most of the major earthquakes of the last 50 years

(Berberian, 1995; Talebian and Jackson, 2004), along the Mountain Front Flexure (Figs. 1, 2), a major tectonic lineament of the area. However, the instrumental seismic record indicates that this structure had never produced a $M_w > 7$ earthquake in last decades. Identifying the fault or fault segment activated during the seismic event, and defining its dimensional



parameters, is thus essential for the assessment of the seismic hazard (Wells and Coppersmith, 1994).

In seismically active fold and thrust belts (FTBs), where the earthquake dataset is not sufficiently robust to constrain the geometry of active faults, deep cross-sections built

using balancing techniques (Dahlstrom, 1969; Hossack, 1979) have been successfully used to improve the knowledge of the seismogenic structures, as carried out in (e.g.) the Los Angeles area (Shaw and Suppe, 1996; Davis et al., 1989), Taiwan (Yue et al., 2005; Mouthereau and Lacombe, 2006), and the Longmen Shan FTB (Wang et al., 2013). In the Zagros FTB, many of the largest earthquakes are associated with major reverse faults affecting the Precambrian

basement (e.g. Jackson, 1980; Berberian, 1995; Talebian and Jackson, 2004), which are included in almost all the published balanced cross-sections across the belt (Blanc et al., 2003; Molinaro et al., 2005; Mouthereau et al., 2007; Vergés et al., 2011). Despite being located more than 200 km away from the epicentral area, these cross sections suggest that the seismogenic structure of the $M_w$ 7.3 earthquake could be related with the Mountain Front

Flexure, which extends across the aftershock area of the November 2017 earthquake (Figs. 1,2). The flexure, across which a marked variation of both topography and structural relief occurs (Falcon, 1961), is commonly interpreted as produced by a large underlying basement thrust, namely the Mountain Front Fault. This structure thus candidates as the seismogenic fault of the recent $M_w$ 7.3 earthquake.

Geological observations of faults and folds affecting Meso-Cenozoic rocks exposed in the epicentral area are reported in this study. These observations were integrated with the interpretation of near vertical seismic reflection profiles calibrated with well logs, allowing us to produce a detailed and well-constrained geological cross-section reaching a depth ranging from 2 to 5 km. The section was then completed at depth by using the balancing technique



(e.g. Dahlstrom, 1969; Hossack, 1979). Our results indicate that the November 2017 seismic activity is attributable to the Mountain Front Fault, for which, using the balancing technique, we reconstructed 10 km of cumulative displacement in the hypocentral area.

**Geological background**

The NW-SE striking Zagros mountain belt formed due to the continental collision between the Arabian and Eurasian plates (Vergés et al., 2011; Berberian and King, 1981; Alavi, 1994; 2007; Argand et al., 2005; Mouthereau et al., 2006). The present-day northward motion of Arabia relative to fixed Eurasia is about 2 cm/yr (Vernant et al., 2004). This is

partitioned between right-lateral motion along NE-SW-striking faults and NE-SW oriented shortening (Blanc et al., 2003; Vernant et al., 2004; Talebian and Jackson, 2002; 2004), which in the Zagros belt is about 5-10 mm/yr (Vernant et al., 2004). The belt is bounded to the NE by the Main Recent Fault and Main Zagros Fault (Fig. 1), forming the suture zone that separates terrains derived from the Mesozoic conjugate margins of the Neo-Tethyan

ocean. The Zagros FTB, to the SW of the suture, involves units originally pertaining to the Arabian continental margin (Ziegler, 2001; Blanc et al., 2003; Sepehr and Cosgrove, 2004; Ghasemi and Talbot, 2006; Mouthereau et al., 2012; English et al., 2015). Within the Zagros FTB, the High Zagros Fault, a major structure striking NW-SE, separates the Imbricate Zone to the NE, where intensely faulted and folded units are exposed, from the Folded Belt to the

SW (Blanc et al., 2003; Karim et al., 2011; Vergés et al., 2011). The SW boundary of the Zagros FTB is the Mountain Front Flexure, corresponding to a basement and topographic step that divides the belt from its foreland basin to the SW (Falcon, 1961). The flexure has a sinusoidal shape, defining salients and recesses along the belt. The seismic sequence of the November 2017 earthquake locates at the boundary between two of them, namely the Kirkuk





embayment and the Lurestan arc (Figs 1, 2). Folds and thrusts of the Folded Belt of the

Kirkuk embayment and of the Lurestan arc are NW-SE-striking, becoming locally NNW-

SSE-trending along the boundary between the two domains. There, a major bend of the

Mountain Front Flexure occurs (Vergés et al., 2011; Sadeghi and Yassaghi, 2016) (Figs. 2,3).

Indeed, the envelope of NNW-SSE striking en-echelon folds along the Mountain Front

Flexure in the epicentral area of the November 2017 earthquake roughly runs N-S (Fig. 2).

This is interpreted as being associated with the occurrence of a N-S-striking basement fault

(i.e. the Khanaqin Fault; e.g. Berberian, 1995; Hessami et al., 2001) that should presently act

as a right-lateral fault. Folds in the Lurestan arc affect an about 10 km-thick sedimentary

succession (Hessami et al., 2001; Ziegler, 2001; Homke et al., 2009; Vergés et al., 2011;

English et al., 2015). In detail, the uppermost Proterozoic basement of the Arabian plate in

the Lurestan region is overlain by a nearly 3000 m thick Paleozoic succession dominated by

continental clastic deposits (Jassim and Goff, 2006; Bordenave, 2008). The strong rheological

contrast between the crystalline basement and the overlying sedimentary cover makes the

basement-cover interface a major decollement horizon of the Lurestan region (e.g. Vergés et

al., 2011), despite the lack of evidence for the occurrence of the Hormuz salt at the base of

the sedimentary pile of the study area. Permian rifting, related to the opening of the Neo-

Tethys ocean (Berberian and King, 1981; Sepehr and Cosgrove, 2004; Ghasemi and Talbot,

2006), led to the deposition of about 1 km of shallow-water carbonates (Chia Zairi Fm.)

(Jassim and Goff, 2006; Bordenave, 2008), with at the base some tens of meters of shales,

forming a mobile level sandwiched between two competent packages (Fig. 3). With

continuing passive margin subsidence, nearly 1800 m of Triassic-Lower Jurassic shallow-

marine carbonates and evaporites, with minor shales, accumulated (Mirga Mir to Sekhaniyan

Fm.) (Jassim and Goff, 2006; Bordenave, 2008). This interval is essentially formed by



competent units, with the exception of the about 100 m thick Baluti and Bedu shales Fms., at the top and base of the Triassic succession, respectively. This is a remarkable difference with respect to the Fars and Dezful Embayment areas to the SE of the Zagros Belt, where the dolostones and limestones of the Triassic Kurra Chine Fm. are substituted by the evaporite-

dominated Dashtak Fm., which there acts as a major decollement level. A major late Early to Middle Jurassic subsidence pulse led to carbonate platform drowning and deposition of about 100 m of relatively deep-water limestones, marls and black shales and evaporites (Sargelu, Naokelekan, Barsarin Fm., Toarcian to Tithonian), followed by 700 m of Cretaceous basinal limestones, shales and marls (Garau, Sarvak and Ilam Fms) (Jassim and Goff, 2006;

Bordenave, 2008). The closure of the Neo-Tethys Ocean during the Late Cretaceous led to the formation of a flexural basin, filled by a ca. 2 km thick Maastrichtian to Eocene succession (Hessami et al., 2001; Homke et al., 2009; Vergés et al., 2011; Saura et al., 2015), evolving from deep-marine marls and limestones to a prograding wedge of deep marine to continental clastic sediments. This first foredeep infill is overlain by about 500 m of shallow-

water carbonates of the Shahbazan and Asmari Fm (Oligocene-lower Miocene), passing upward to lower Miocene evaporites. Renewed shortening and thrusting from the late Miocene to the recent led to the deposition of a younger foreland basin clastic infill (Fig. 3) (Hessami et al., 2001; Jassim and Goff, 2006; Homke et al., 2009).

**NE-SW geological cross-section**

In this paragraph we present a NW-SE-oriented geological section across the study area. The section is divided into two portions. Figures 4 and 5 illustrate the NE and SW portion of the section, respectively (with a small overlap area). Two seismic reflection profiles running at a low angle to the geological cross-section trace are projected onto the



section plane, and key field observations along the NE portion of the section are also reported in figure 4.

The High Zagros Fault to the NE of the study area intersects the cross section of figure 4 in its northern portion. There, the major thrust fault dips roughly parallel to the strata of both hanging-wall and footwall blocks (i.e the cutoff angles are close to zero). Cretaceous strata in the footwall are affected by the NW-SE striking, tens of km-long thrusts of the Satiary Thrust System. These thrusts have low (< 10°) hanging-wall and footwall cutoff angles (Fig. 4). Along the section, the Garau Fm. sits in the hanging wall of the thrust and the Ilam Fm. lies in its footwall. However, the geological map of figure 3 shows that the Sehkanian Fm. is the oldest exposed unit in the hanging-wall block and that it is thrust on top of the Upper Cretaceous Gurpi Fm. (see also the field photograph of figure 4), which lies about 1000 m higher in the stratigraphic column. This feature, coupled with the observed hanging-wall flat on footwall flat relationship, suggests displacements in the order of several kilometres. In the footwall of the Satiary thrust system, Upper Triassic to Cretaceous strata are, as a whole, 20-30° NE-dipping for about 4 km, until they meet the tens of kilometres long Herta Thrust System. This includes two 30°-dipping thrusts (joining SE-ward; Fig. 3) showing very low cutoff angles and separating the Triassic Sarki Fm. in the hanging wall of the trailing thrust from the Sargelu and Garau fms. in its footwall (Fig. 4). The repetition of hanging-wall flat on footwall flat geometries (Fig. 4) indicates a remarkable (i.e. several kilometres) displacement also for the Herta Thrust system.

Near-vertical reflection seismic profiles in this northern area are affected by a significant noise; however, both the Satiary and the Herta thrust systems are imaged at depth (Fig. 4) displaying very low cutoff angles, which confirms their significant horizontal displacement. Folds associated with the Herta and Satiary thrust systems are truncated by the High Zagros





Fault in the SE portion of the study area. This may be observed in the eastern portion of the geological map of figure 3 and, more in detail, in the photograph of figure 4, where the sub-horizontal High Zagros Fault truncates an anticline exposing the Gurpi Fm. in the limbs and the Ilam Fm. in the core. This observation constrains the relative timing of development of

these structures, pointing to an out of sequence emplacement (or reactivation) of the High Zagros Fault, which post-dates the development of the Herta and Satiary fault systems. Moving to the southwest, the Marakhil Anticline exposes the Geli Khana Fm. in its core, and the seismic profile indicates that the Paleozoic strata are folded as well. The Marakhil Fault, bounding the anticline to the SW, has a high (> 60°) hanging-wall cutoff angle, typical of a

reactivated (i.e. positively inverted) extensional fault (e.g. Sibson, 1985; Williams et al., 1989). The fault flanks to the NE an about 5 km-wide gentle syncline affected by low-displacement (i.e. < 100 m) reverse faults with both low (e.g. the Qlaji Thrust) and high (e.g. the Bawrol Thrust) cutoff angles. In detail, similarly to the Marakhil Fault, the Bawrol Thrust has a hanging-wall cutoff angle typical of a positively inverted normal fault, whose original

extensional activity  post-dated the deposition of the Sehkaniyan Fm.. Indeed, syn-kinematic thickening of the Sargelu, Naokelekan, and Barsarin formations (S-N-B in Fig. 4) observed across the Marzan extensional fault, as well as wedging of the same formations in the hanging wall of the Qlaji Thrust, indicate that many of the previously illustrated inverted faults (affecting Triassic and Jurassic strata), developed during a Middle Jurassic extensional

pulse. The Sheykh Saleh Anticline is another major structure of this part of the Lurestan region. It separates an area to the SW, where the oldest rocks exposed in the cores of the anticlines (Gheytuleh, Azgaleh, and Miringeh anticlines) belong to the Upper Cretaceous Ilam Fm., from an area to the NE where the oldest rocks exposed at the core of the anticlines belong to the Triassic Kurra Chine and Geli Khana Fms (Fig. 3). The NE block has a





structural relief of about 2 km. Despite the significant noise affecting the seismic section, the Ilam and Sehkaniyan Fms. are clearly imaged in the subsurface of the area SE of the Sheykh Saleh Anticline (Fig. 5). Both formations are made of carbonates and are capped by shales and marls of the Sargelu and Gurpi Fms., respectively, this making their top strongly

reflective and recognisable. The first clear occurrence of the top Sehkaniyan reflectors is underneath the southwestern limb of the Gheytuleh Anticline, at about 1 s TWT (Fig. 5), entirely consistent with the dip and thickness of the overlying stratigraphic units. These Sehkaniyan reflectors are SW-dipping and become NE-dipping about 2 km to the SW, below the syncline flanking to the SW the Gheytuleh Anticline. This coherence between surface and

subsurface geometries points to a roughly parallel folding of the entire package overlying the Sehkaniyan Fm. About 1 km to the SW, also the top Ilam reflectors become recognisable. Further to the SW, starting from the Azgaleh Anticline area, reflectors are calibrated with well logs and exposures of the top Ilam Fm. In this southwestern portion of the section, the envelope of the top of the Ilam and Sehkaniyan formations defines a 2-5° SW-dipping,

regional-scale panel, with limited decoupled deformation between the Mesozoic and Cenozoic units due to the occurrence of a weak package comprised between the stiff Ilam and Asmari Fms. This shallow-dipping faulted and folded panel terminates at the Miringeh Anticline, which displays an unfaulted forelimb. There the strata of the entire Paleozoic to Cenozoic sedimentary succession are parallel and form a 10 km wide SW-dipping monocline.

This latter is bounded by two N-S striking anticlines cored by the Asmari Fm.; below them, a repetition of the Mesozoic reflectors is observed, which is produced by a backthrust. At the SW termination of the seismic sections, the entire Paleozoic to Cenozoic sedimentary succession becomes horizontal and forms a large-scale syncline.



**Balancing the cross-section**

The cross-section shown in figures 4 and 5 is completed at depth by producing a geological

solution (Fig. 6) in which line-length preservation during folding and thrusting is assumed

(e.g. Dahlstrom, 1969; Hossack, 1979). The balanced cross-section is built along a direction

oriented N49°, which is perpendicular to the trend of major folds and thrusts. These

structures display negligible regional plunge along the section, which allows us to use a

vertical plane to build the section. This also ensures the absence of remarkable out-of-plane

motion and allows us to directly compute the thickness of the exposed Mesozoic and

Cenozoic units along the section. The chosen section plane forms an angle of 17° with the

N215°-striking and 78° dipping plane containing the P and T axes of the of the 2017 $M_w$ 7.3

earthquake, thus representing a proper section to obtain insights on the seismogenic

structures.

Some lateral thickness variations, in the order of some tens of metres, are observed

for the package comprised between the Sargelu and Barsarin fms.. The Sehkaniyan and Sarki

fms. also display lateral thickness variations of the same order of magnitude. In the Geli

Khana and Kurra Chine fms. we have not observed any kind of growth structure, and the

parallelism between reflectors observed in the seismic line of figure 4 indicates that the

thickness of these formations can be considered roughly constant. These observations

indicate that, as a whole, a constant thickness can be used for the almost 2 km thick package

comprised between the base of the Geli Khana Fm. and the base of the Garau Fm.. The

overlying units are not continuously exposed in the northern part of the section and, because

of that, they are not shown in the restoration. The Paleozoic units and the basement, for

which only limited and discontinuous information is available, are modelled using 1 km and

km thick layers, respectively. For the sake of simplicity, thickness variations in Upper





Paleozoic units are firstly neglected and then re-introduced after cross-section balancing. This because the adoption of constant thickness for the entire upper crust and of flexural slip folding allowed us to assume line-length preservation. Coherently, the restored cross section shows the cumulative length of Mesozoic, Paleozoic, and basement layers. The trace of the

faults in the restored section is obtained by smoothing the polyline built by connecting the restored cutoff points. This is done to avoid zig-zag effects and, in any case, smoothing is less than 0.5% of the original cutoff point position.

Coherently with field observation, in our reconstruction thrusts to the NE of the Marakhil Anticline are thin-skinned and have a displacement in the order of some kilometres.

They splay off from a basal decollement located at the bottom of the Triassic sequence, namely within the Bedu Shale, sandwiched between the competent Chia Zairi and Geli Khana-Kurra Chine packages. The Marakhil Anticline is instead a deeply rooted structure, associated with the Marakhil inverted normal fault, which is observed at the surface (Fig. 4). The simple shallow geometry of this large wavelength fold introduces a geometrical problem

at depth, as two solutions can be applied to model the deeper portion of the anticline. In the first one, the inverted fault affects only the sedimentary cover, the core of the anticline is filled by ductile material and the underlying basement is not involved in faulting and folding. In the second solution, the inverted fault involves also the basement. The lack of a sufficiently thick ductile layer at the base of the Paleozoic sequence, and the occurrence of a

structural step across the Marakhil Anticline, are more compatible with the second, basement-involved, solution. Following this structural model, and keeping constant the line-length of both basement and  cover, we solved the geometry in the core of the anticline by assuming the occurrence of a footwall shortcut of the inverted normal faults in the basement. This represents a typical feature associated with the inversion of normal faults (e.g. McClay,





1989). In our solution, this shortcut transfers displacement from the main reactivated fault to the base of the sedimentary cover. Low-displacement, SW-verging reverse faults and a major back-thrust accommodate such a displacement in the Mesozoic and Paleozoic strata, respectively. The Sheykh Saleh Anticline to the SW shows a similar deep structure, which is

even better supported by the remarkable structural step occurring at this location. Here, a positively inverted normal fault with a footwall shortcut occurs in the basement. The footwall shortcut transfers displacement from the main reactivated fault to the base of the sedimentary cover sequence. Such a displacement is accommodated by folding and faulting of the sedimentary cover, with the Paleozoic or Lower Triassic incompetent units (i.e. the Bedu

Shale Fm or the shaly level at the base of the Chia Zairi Fm.) promoting decoupling between Mesozoic and Paleozoic strata. In our interpretation, a positively inverted normal fault bounds to the NE the Miringeh Anticline too, producing the uplift of the crustal block in its hanging wall and preventing the southward propagation of the deformation of the sedimentary cover. Indeed, Paleozoic to Cenozoic strata in the crest and in the wide,

homogeneously dipping SE limb of this anticline are parallel, unfolded and unfaulted. This limb is underlain by a basement low-angle thrust, corresponding to the Mountain Front Fault, on which the main shock is located (Fig. 6). The focal mechanism provided by the USGS indicates a 351°-striking and 16° dipping thrust fault, whose intersection with our N49°-striking vertical section gives 14° of apparent dip. Coherently, in our reconstruction the thrust

dips 15° at the hypocentral depth and becomes almost sub-horizontal upward, where it reactivates the basement-cover interface. A back-thrust splays from this upper flat, accommodating part of the displacement transferred from the main ramp of the Mountain Front Fault, and forming together with it a fishtail structure responsible for the surface deformation observed from interferometric data.

An independent quality check of our reconstruction is provided by the top of magnetic basement data (Fig. 6), whose top is computed according to the regional depth map in Teknik and Ghods (2017). The depths of the crystalline basement underlying the sedimentary cover and the top of the magnetic basement obviously do not coincide, due to the

heterogeneous nature of the magnetic basement. However, their large-scale shape is similar, confirming the occurrence of highs and lows predicted by our reconstruction. The restored length of the section is 104 km, with a negligible maximum error of 1.5%. The total shortening is 20 km, 8 km of which being associated to the thin-skinned Satiary and Herta thrust systems to the NE of the Marakhil Anticline. As previously mentioned, these thrusts

are truncated by the High Zagros Fault, which in this area was active during the Late Cretaceous to Paleocene interval (Karim et al., 2011; Vergés et al., 2011; Saura et al., 2015). These thrusts have also anomalously high displacements compared to the other structures along the section. For both reasons, the Satiary and Herta thrust systems are interpretable as footwall splays of the High Zagros Fault, probably merging with it to the NE, outside the

section. Lower displacements are instead associated with the Marakhil (2.5 km), Sheykh Saleh (2.0), and Miringeh (1.0) faults, the amount of shortening accommodated in the area between the Marakhil and Miringeh anticlines being 5.3 km. The remaining shortening is accommodated by the Mountain Front Fault and associated structures.

**Discussion**

According to our reconstruction, the Mountain Front Fault has 9.7 km of cumulative displacement at 20 km depth, where the main shock nucleated. The displacement decreases upward, becoming 5.8 km at the upper flat. About 1 km of this is accommodated by the frontal back-thrust, while 4.3 km of shortening is transferred to the foreland structures to the

SW of our balanced-cross section. The computed 9.7 km of displacement of the Mountain

Front Fault at the hypocentre are broadly consistent with the 13 km proposed for the same

structures 200 km to the SE (Blanc et al., 2003; Vergés et al., 2011). The earthquakes of the

November 2017 seismic sequence can thus be attributed to movement of the Mountain Front

5    Fault, which forms part of a thrust system splaying from a mid-crustal decollement (Vergés et

al., 2011), similar to that documented in other FTBs (Cristallini and Ramos, 2000; Butler et

al., 2004; Lacombe and Bellahsen, 2016). The important occurrence of reactivated

extensional faults documented in this study suggests that the mid-crustal decollement could

represent a reactivated inherited extensional decollement (e.g. Marshak et al., 2000; Tavani.,

10    2012).

Interferometric data show that the maximum surface deformation occurs at the SW

edge of the geological section (Fig. 6). This reveals that the coseismic displacement has

induced slip along the shallower, near horizontal, upper flat located 20 km to the SW of the

main shock, at the basement-cover interface. Decoupling between the Mesozoic and

15    Paleozoic successions, and between Paleozoic strata and the basement, has strong

implications in terms of seismic potential. As already pointed out by Nissen et al. (2011),

decoupling at the base of the cover sequence implies vertically confined faults, with down-

dip width smaller than 8 km. In fact, only four faults affect the entire upper crust: the three

major steeply-dipping inverted normal faults splaying out from the basal decollement,

20    probably corresponding to the brittle-ductile transition, and the Mountain Front Fault. The

former ones, with their cross-sectional length of up to 25 km, can generate a down-dip

rupture width exceeding 8 km, required for an $M_w$ 6 earthquake (Wells and Coppersmith,

1994). On the other hand, the Mountain Front Fault is the only fault on which a down-dip

rupture width of 30 km, required for an $M_w$ 7.3 earthquake, may occur.



Beyond their importance for seismic hazard assessments, the data illustrated in this work have major implications in terms of a better understanding of thrust tectonics in the Zagros Mountains. The occurrence of salients and recesses is a common feature in fold and thrust belts (Marshak, 1988) including the Zagros, where different mechanisms are invoked

to explain the occurrence of bends in the trace of the Mountain Front Fault (Malekzade et al., 2016, and references therein). According to the scaling relationship of magnitude vs. rupture area (Wells and Coppersmith, 1994), the rupture area for the Iran-Iraq $M_w$ 7.3 earthquake should exceed $10^3$ km$^2$. Therefore, the low-angle Mountain Front Fault must extend in the area where the Mountain Front Flexure runs roughly N-S (Fig. 2). This, coupled with the N-S

clustering of aftershocks (Fig. 2) triggered by SE directed co-seismic slip along the low angle thrust ramp, clearly points to the occurrence of a lateral ramp beneath the N-S segment of the Mountain Front Flexure at the boundary between the Kyrkuk embayment and the Lurestan arc. A further implication of our work concerns the role structural inheritance in the Zagros FTB. The age of rifting and passive margin development is still a matter of debate in the

tectonic puzzle of the area. A Permian to Early Triassic age is commonly inferred for the onset of rifting in the Zagros area (e.g. Berberian and King, 1981; Ghasemi and Talbot, 2006). However, we observed extensional structures that developed synchronously with the deposition of the Middle Jurassic Sargelu Fm., the Marzan extensional Fault (Fig. 4) being the most striking one. The positively inverted Marakhil and Bawrol faults, affecting Upper

Triassic and Lower Jurassic units (thus younger than the main rifting event) also fit well into an Early to Middle Jurassic extensional episode. Such an extensional pulse could also explain the drowning of the long-lived Triassic-Jurassic carbonate platform and the onset of deep-water conditions in the area (Ziegler, 2001; Jassim and Goff, 2006; Bordenave, 2008). Accordingly, for many of the inverted basement extensional faults, a polyphase extensional

history could be proposed, including a Permo-Triassic development and a Middle Jurassic extensional reactivation.

**Conclusions**

The integration of field data, near vertical seismic reflection profiles, and earthquake data, allowed us to provide a comprehensive picture of the geometry and dimensional parameters of the faults in the hypocentral area of November 2017 seismic sequence at the Iran-Iraq border. The tectonic framework of this area includes a mid-crustal decollement level at a depth of ca. 20 km, from which high angle positively inverted normal faults splay

off. At its southwestern edge, the decollement ramps up, to form the Mountain Front Fault, which joins southward an upper decollement level located at the basement-cover interface. The occurrence of multiple decollement levels in the sedimentary succession promotes a partly decoupled deformation, and limits the size of most of the faults of the area. The main shock of the November 2017 $M_w$ 7.3 earthquake nucleated in the basement, along the

Mountain Front Fault. Co-seismic slip unzipped the shallower portion of the fault to the SW, at the basement-cover interface, and activated structures responsible for the observed surface deformation.

**Acknowledgments**

We acknowledge the use of imagery from the Land Atmosphere Near-real time Capability for EOS (LANCE) system, operated by the NASA/GSFC Earth Science Data and Information System (ESDIS) with funding provided by NASA/HQ, and of Copernicus Sentinel data 2017, processed by ESA. The geological cross sections presented in this work were constructed



using the Midland Valley 3D Move software. Requests for obtaining the near vertical seismic

sections and wells  data should be submitted to the National Iranian Oil Company.



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





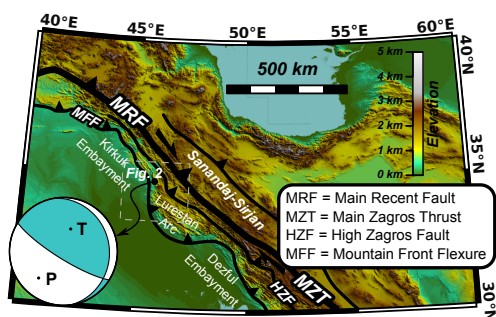

**Figure 1** (Single column)
Tectonic sketch map of the Zagros Mts., showing epicenter and moment tensor of the November 12, 2017 Mw 7.3 earthquake (source USGS, https://earthquake.usgs.gov/)




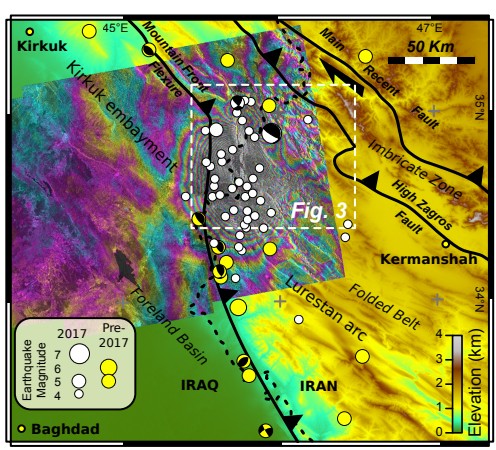

**Figure 2** (Single column)

Elevation map (source ESDIS) showing the main structural features of the Lurestan region and earthquake distribution (source USGS, https://earthquake.usgs.gov/). Mw > 4 earthquakes of the November 2017 sequence are reported in white; pre-2017 Mw > 5 earthquakes are reported in yellow. The Sentinel 1 co-seismic interferogram (Nov. 11, 2017, 3 p.m. UTC to Nov. 17, 2017, 2:59 p.m. UTC; http://sarviews-hazards.alaska.edu/Event/34/) is also shown as an overlay.





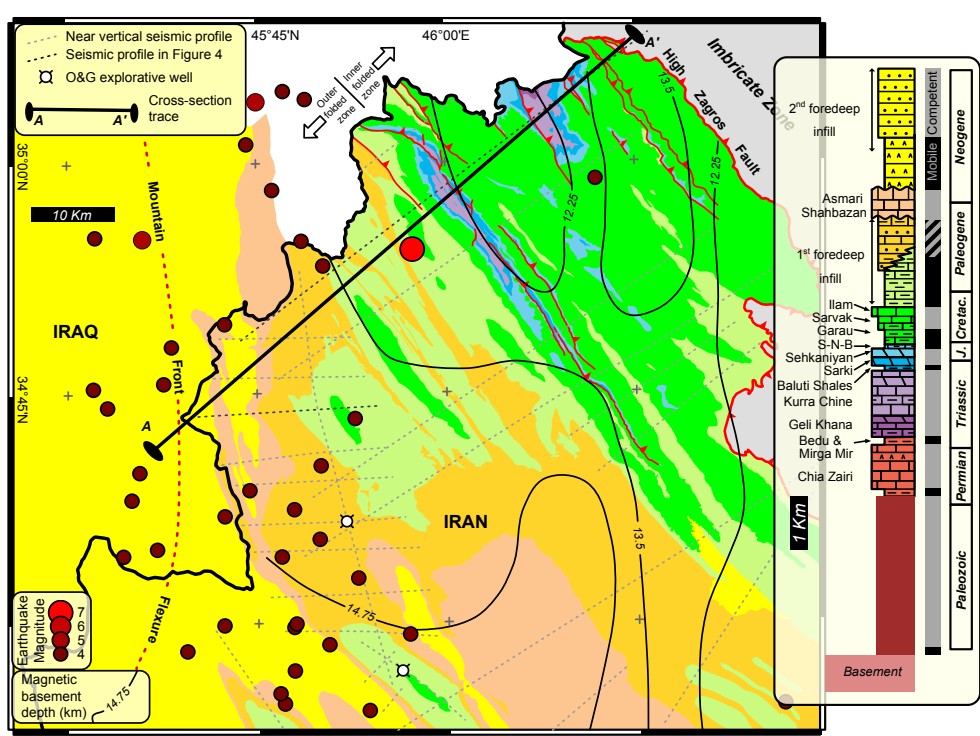

**Figure 3** (Double column)

Geological map of the NW portion of the Lurestan region (source: National Iranian Oil Company and original field mapping) showing: (i) November 2017 earthquakes; (ii) traces of near vertical seismic sections and wells used to constrain the geological cross-section of figure 6 (sections shown in figures 4 and 5 are in black ); (iii) magnetic basement depth (Teknik and Ghods, 2017), and (iv) trace of the section in figures 4 and 5. The inset shows the stratigraphic succession of the area, with thicknesses for the Mesozoic to Cenozoic stratigraphic units computed from original field data. Thickness for the Paleozoic to Lower Triassic is taken from the literature on the geology of Iraq (Jassim and Goff, 2006)





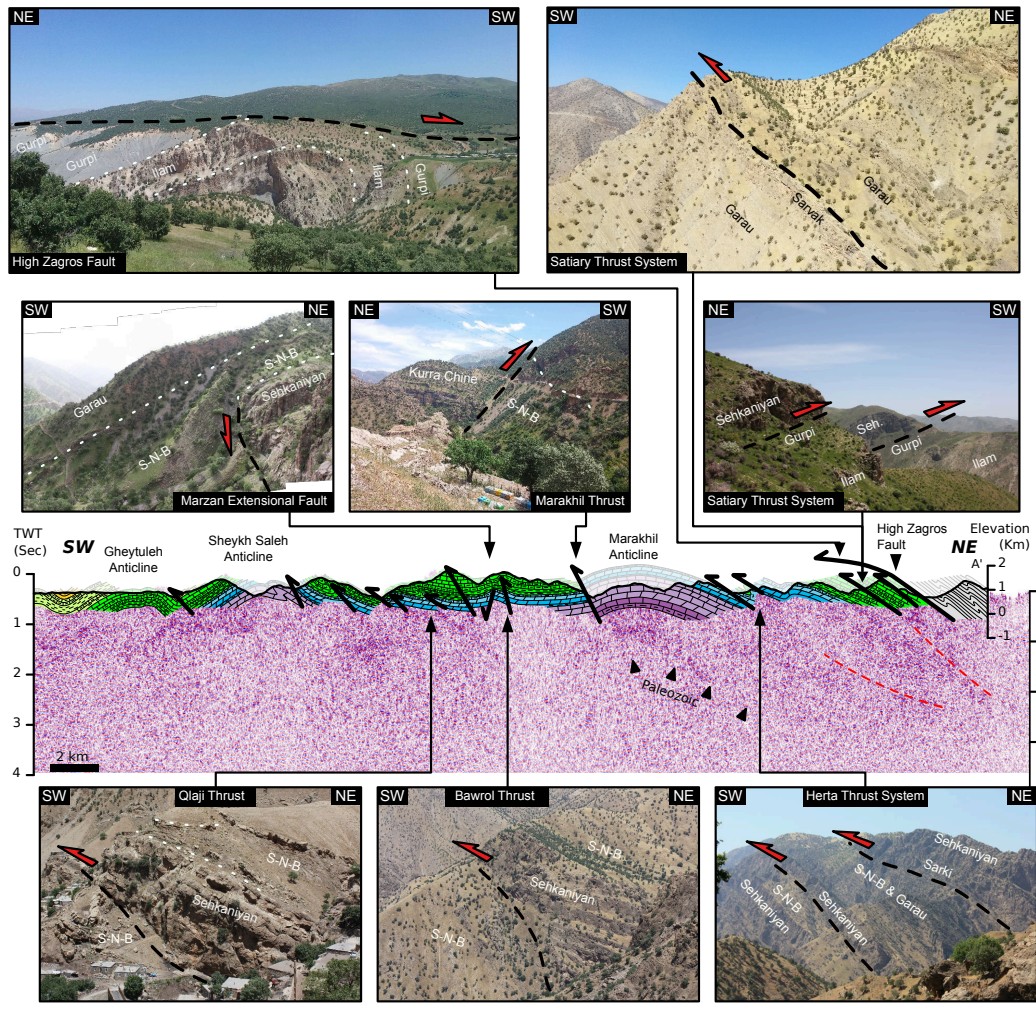

**Figure 4** (Double column)
NE part of the NE-SW oriented geological section across the hypocentral area, with field photographs illustrating the main structural features. A near vertical seismic profile is displayed below the cross section (vertical scale is roughly equal to the horizontal scale).



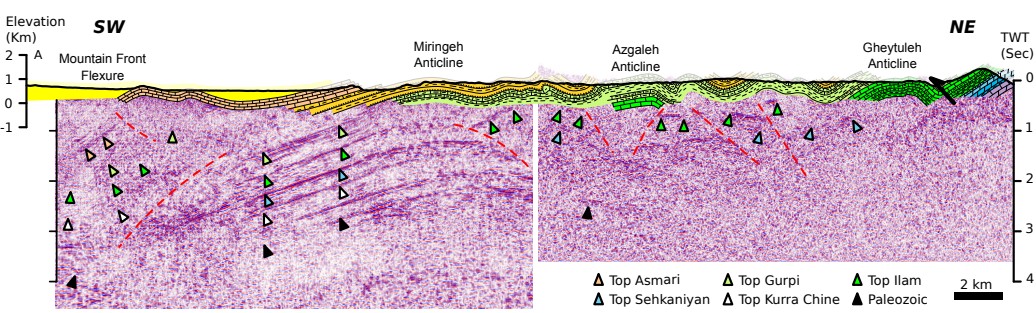

**Figure** 5 (Double column)

SW part of the NE-SW oriented geological section across the hypocentral area. Near vertical seismic profiles are displayed below the cross section (vertical scale is roughly equal to the horizontal scale).



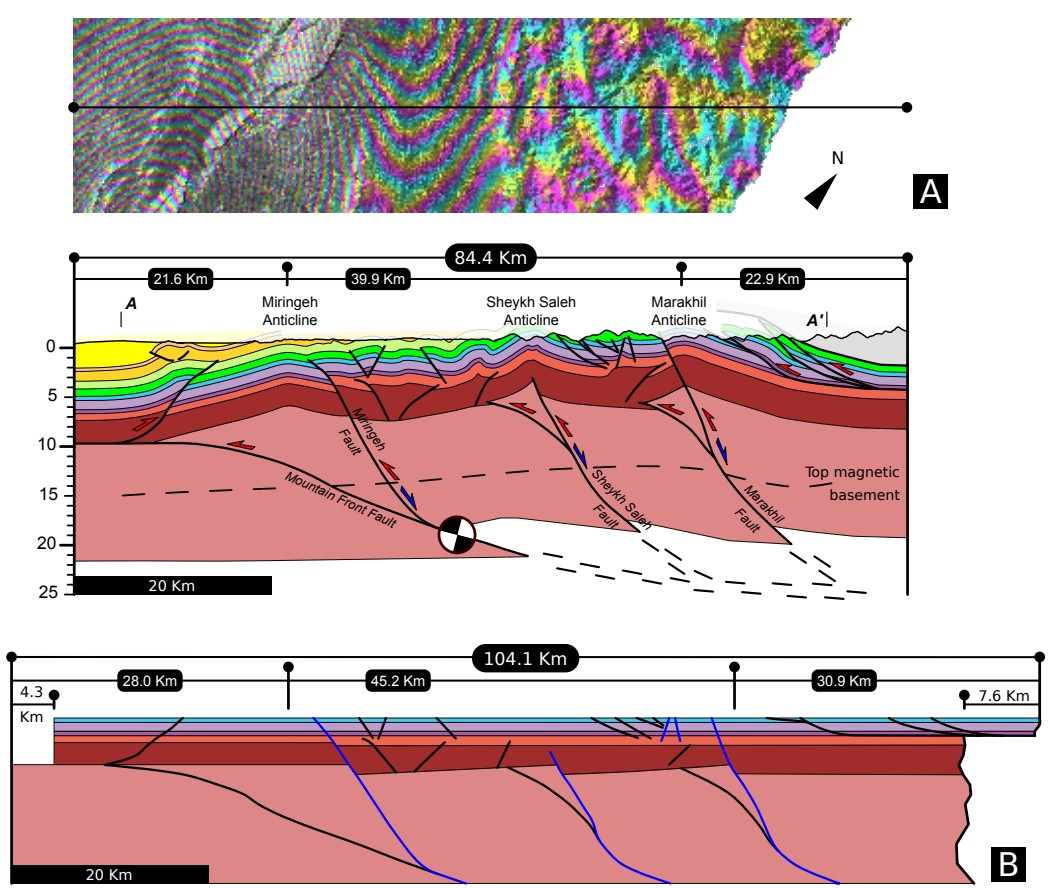

**Figure 6** (Double column)
(A) Balanced cross-section along the direction of the section in figures 4 and 5, showing projected main shock and detail of the co-seismic interferogram with trace of the section. (B) Restored section.