# Peer review of "The seismogenic fault system of the 2017 $M_w$ 7.3 Iran-Iraq earthquake: constraints from surface and subsurface data, cross-section balancing and restoration"

_Solid Earth, 2018_

## Referee Comment (RC1) · Anonymous Referee #1 · 22 Mar 2018

Evaluation :

1. Does the paper address relevant scientific questions within the scope of SE? YES 2. Does the paper present novel concepts, ideas, tools, or data? YES 3. Are substantial conclusions reached? YES 4. Are the scientific methods and assumptions valid and clearly outlined? YES 5. Are the results sufficient to support the interpretations and conclusions? YES 6. Is the description of experiments and calculations sufficiently complete and precise to allow their reproduction by fellow scientists (traceability of results)? N/A 7. Do the authors give proper credit to related work and clearly indicate

their own new/original contribution? YES 8. Does the title clearly reflect the contents of the paper? YES 9. Does the abstract provide a concise and complete summary? YES 10. Is the overall presentation well structured and clear? YES 11. Is the language fluent and precise? YES. Some mistakes to be fixed 12. Are mathematical formulae, symbols, abbreviations, and units correctly defined and used? N/A 13. Should any parts of the paper (text, formulae, figures, tables) be clarified, reduced, combined, or eliminated? NO 14. Are the number and quality of references appropriate? YES. One reference to be added for fair acknowledment of previous work 15. Is the amount and quality of supplementary material appropriate? N/A

Formal review :

This is the second time I act as reviewer for this ms by Tavani et al. that reports what is to me a timely structural analysis of the fault system that is likely the locus of the 2017 Mw 7.3 Iran-Iraq earthquake. The manuscript is well written and nicely illustrated and to me it provides useful insights into the characterization of the fault that presumably ruptured during the earthquake. Most of the many comments I made in my former review have been included in this new version that, I find, has significantly improved.

I therefore recommend acceptance after some minor revision (see below).

-Plot the striation corresponding to the co-seismic slip line deduced from the focal mechanism on the stereoplot of Fig.1

-Add orientation of the section on Figure 6

-P8 L14, P12 L18, P13 L2 : change whose (for person) into which (for object)

- P14 L6-7 : it would be fair to acknowledge, hence to cite here the paper by Lacombe and Mouthereau Tectonics, 2002 entitled : Basement-involved shortening and deep detachment tectonics in forelands of orogens and which therefore fully meets the topic of crustal decollement in deformed forelands.

-P15 L10 : SW instead of SE ?

-P15-16 : The age of the inherited normal faults is now discussed in more detail than in the previous version, but still fails to explain thickness variation of pre-Permian Paleozoic strata.

P16 L8 : I would change into 'includes a likely mid-crustal decollement'

––––––––––––––––––––––––––

---

## Author Comment (AC1) · 27 Apr 2018

Comment. Plot the striation corresponding to the co-seismic slip line deduced from the focal mechanism on the stereoplot of Fig.1 Response. OK

Comment. Add orientation of the section on Figure 6 Response. OK

Comment. P8 L14, P12 L18, P13 L2 : change whose (for person) into which (for object) Response. OK

[Figure]

Comment. P14 L6-7 : it would be fair to acknowledge, hence to cite here the paper by Lacombe and Mouthereau Tectonics, 2002 entitled : Basement-involved shortening and deep detachment tectonics in forelands of orogens and which therefore fully meets the topic of crustal decollement in deformed forelands. Response. OK

Comment. P15 L10 : SW instead of SE ? Response. Yes, it is SW

Comment. P15-16 : The age of the inherited normal faults is now discussed in more detail than in the previous version, but still fails to explain thickness variation of pre-Permian Paleozoic strata. Response. OK. Thickness variation is inferred from a mid-Paleozoic unconformity seen in some seismic sections, including that in figure 5 (below the Miringeh Anticline, at a depth of 2.6-3 seconds). This will be indicated in the new version of figure 5 and it will be mentioned in the text

Comment. P16 L8 : I would change into 'includes a likely mid-crustal decollement' Response. OK

―――――――――――――――――――――

---

## Referee Comment (RC2) · Anonymous Referee #2 · 16 May 2018

This is a nice structural reconstruction of the western Zagros, integrating 2017 earthquake data. The main issue is a fixation on the earthquake taking place on a N-S segment of the MFF (Mountain Front Fault). Brief mention is made of the N-S structure being separate – the Khanaqin Fault, but this is then strangely ignored. In fact, it looks very likely that the earthquake took place on the Khanaqin Fault – and is distinct from NW-SE fault segments grouped as the MFF. This is a significant aspect of the regional geology, which should be emphasised rather than underplayed.

The root of the problem is that the Zagros faults get depicted in different ways. One

view is to emphasise their continuity, so that the MFF, HZF etc get drawn as continuous structures over 100s of km (see Berberian et al 1995). If these faults are offset by N-S right-lateral faults, the offsets are sometimes depicted as up to 100s of km (see Berberian again), but more detailed work shows that such offsets are only a few km (Authemayou et al., 2006). However, the faults are much more segmented than this "Himalayan" style – see work by Walker, Ramsey et al, with segments typically no more than 20-40 km, rupturing in M 5-6 earthquakes. The fault segments linked together as the "MFF" are not a Himalayan-style nappe, but equivalent steps in the relief and geomorphology of the range. Therefore the Tavani et al paper needs to consider the consequences of the N-S Khanaqin Fault being a separate, N-S structure to the main NW-See thrusts, which slipped in the 2017 earthquake in a highly unusual manner for the Zagros – witness the sheer size of the event, which is much larger than typical Zagros thrust earthquakes. . See Lawa et al (2013) and Allen et al (2013) for examples of Zagros structure maps that include the Khanaqin Fault.

The geology descriptions and structural sections look very good, but this issue of fault segmentation and the existence of the Khanaqin Fault means that they need more work. The early part of the paper describes the 2017 earthquake parameters, but another way of doing this is to quote the slip vector azimuth of the event, which is 90 deg. from the auxillary plane strike, ie towards 212 deg. by my calculation. This means highly oblique slip on the fault, and also that the section line in figure 6 is covering faults with very different orientations, from the conventional NW-SE thrusts to the more N-S Khanaqin Fault. Neither of these points comes across properly in the paper. It would help if the Khanaqin Fault trace was properly drawn on Figures 2 and 3. The authors seem to have taken the continuous, sinusoidal, lines drawn on many regional papers for the Zagros, but, as noted, there are plenty of other papers that try to draw the Khanaqin Fault more accurately. Where Tavani et al make an improvement on our knowledge is that the try use the 2017 earthquake data to interpret the fault for the first time at depth, as a lateral ramp: this point stands, despite their confusion over the structure being part of the "MFF".

See also Koshnaw et al 2017 for a cross-border geology map that means figure 3 can be improved.

A lat/long label in fig 3 should be 45/45 E not 45/45 N.

Page 3: This structure is thus a candidate...

The first part of p 15 is critical, as the authors make a good description of the likely regional structure - but this is not apparent on their maps or cross-sections!

---

## Author Comment (AC2) · 17 May 2018

Point 1 This is a nice structural reconstruction of the western Zagros, integrating 2017 earthquake data. The main issue is a fixation on the earthquake taking place on a N-S segment of the MFF (Mountain Front Fault). Brief mention is made of the N-S structure being separate – the Khanaqin Fault, but this is then strangely ignored. In fact, it looks very likely that the earthquake took place on the Khanaqin Fault – and is distinct from NW-SE fault segments grouped as the MFF. This is a significant aspect of the regional geology, which should be emphasised rather than underplayed. Response

The main-shock and the Khanaqin fault are about 20 km apart. We will show this in figure 3.

Point 2 The root of the problem is that the Zagros faults get depicted in different ways. One view is to emphasise their continuity, so that the MFF, HZF etc get drawn as continuous structures over 100s of km (see Berberian et al 1995). If these faults are offset by N-S right-lateral faults, the offsets are sometimes depicted as up to 100s of km (see Berberian again), but more detailed work shows that such offsets are only a few km (Authemayou et al., 2006). However, the faults are much more segmented than this "Himalayan" style – see work by Walker, Ramsey et al, with segments typically no more than 20-40 km, rupturing in M 5-6 earthquakes. The fault segments linked together as the "MFF" are not a Himalayan-style nappe, but equivalent steps in the relief and geomorphology of the range. Response We did not make this part very clear. We have never supported the idea that the mountain front fault is a continuous structure. Indeed, what we have drawn in figures 1, 2, and 3 is the trace of the mountain front flexure. However, in figures 1 and 2 we have used for the flexure the same pattern as for the faults, and this has created some misunderstanding. Figures 1 and 2 will be modified accordingly.

Point 3 Therefore the Tavani et al paper needs to consider the consequences of the N-S Khanaqin Fault being a separate, N-S structure to the main NW-SE thrusts, which slipped in the 2017 earthquake in a highly unusual manner for the Zagros – witness the sheer size of the event, which is much larger than typical Zagros thrust earthquakes. See Lawa et al (2013) and Allen et al (2013) for examples of Zagros structure maps that include the Khanaqin Fault. The geology descriptions and structural sections look very good, but this issue of fault segmentation and the existence of the Khanaqin Fault means that they need more work. Response We will add the trace of the Khanaqin fault in figure 3. We will also remark that: (1) the Khanaqin fault cannot be the source of the 7.3 earthquake (see point 1), (2) this fault coincides with the backthrust seen at the SW termination of the section (figures 5 and 6).

Point 4 The early part of the paper describes the 2017 earthquake parameters, but another way of doing this is to quote the slip vector azimuth of the event, which is 90 deg. from the auxillary plane strike, ie towards 212 deg. by my calculation. This means highly oblique slip on the fault, and also that the section line in figure 6 is covering faults with very different orientations, from the conventional NW-SE thrusts to the more N-S Khanaqin Fault. Neither of these points comes across properly in the paper. Response The slip vector is more precisely the plane containing the T and P axes, which is also perpendicular to the two nodal planes. This is 215° striking and 78° dipping. As quoted at page 10, the orientation of the section is N49°, i.e. at 14° with respect to the co-seismic slip direction. This direction was chosen because balanced cross-sections must run parallel to the tectonic transport direction, this to ensure the absence of out of plane motion (as quoted in the text). This is a well-established procedure and does not need any further clarification. Concerning the fact that the strike of the nodal plane is oblique to the trend of our section, this is merely because the low-dipping fault is a lateral ramp, and cross-section along lateral ramps must run parallel to the transport direction. Also this basic principle does not need clarification in the text. Concerning the fact that the section runs oblique to the N-S Khanaqin fault, we remark that if this fault exist, it is a second order accommodation structure, and the section must run perpendicular to the main structures.

Point 5 It would help if the Khanaqin Fault trace was properly drawn on Figures 2 and 3. The authors seem to have taken the continuous, sinusoidal, lines drawn on many regional papers for the Zagros, but, as noted, there are plenty of other papers that try to draw the Khanaqin Fault more accurately. Response Done. Added on figure 3.

Point 6 Where Tavani et al make an improvement on our knowledge is that the try use the 2017 earthquake data to interpret the fault for the first time at depth, as a lateral ramp: this point stands, despite their confusion over the structure being part of the "MFF". See also Koshnaw et al 2017 for a cross-border geology map that means figure 3 can be improved. Response We will quote Koshnaw et al 2017.

Point 7 A lat/long label in fig 3 should be 45/45 E not 45/45 N. Response Done.

Point 8 Page 3: This structure is thus a candidate... Response Done.

Point 9 The first part of p 15 is critical, as the authors make a good description of the likely regional structure - but this is not apparent on their maps or cross-sections! Response We now explained our view on the N-S striking Khanaqin Fault. If this fault exist, it is the backthrust imaged at the SW edge of the seismic line in figure 5. Accordingly, we have added this at page 13 (Balancing the cross section) "The position of such a back-thrust roughly coincides with the Khanaqin Fault (e.g. Lawa et al., 2013) (Fig. 3), which accordingly must be downgraded to accommodation structure of the Mountain Front Fault"

In the discussion, at page 15, we have added: "As previously mentioned, the N-S striking Khanaqin Fault (e.g. Berberian, 1995; Hessami et al., 2001; Lawa et al., 2013; Allen et al., 2013), in our structural reconstruction becomes an accommodation structure of the Mountain Front Fault."

The back thrust is also labelled Khanaqin Fault in figure 6.

Please also note the supplement to this comment:
https://www.solid-earth-discuss.net/se-2018-21/se-2018-21-AC2-supplement.pdf

[revised manuscript text omitted]

Allen, M. B., & Talebian, M. (2011). Structural variation along the Zagros and the nature of the Dezful Embayment. Geological Magazine, 148, 911-924. DOI: 10.1017/S0016756811000318

Allen, M. B., Saville, C., Blanc, E. P., Talebian, M., Nissen, E. (2013). Orogenic plateau growth: Expansion of the Turkish-Iranian Plateau across the Zagros fold-and-thrust belt. Tectonics, 32, 171-190. DOI: 10.1002/tect.20025

Bahroudi, A, Koyi, H.A. (2003). Effect of spatial distribution of Hormuz salt on deformation style in the Zagros fold and Thrust Belt: An analogue modelling approach. Journal of the Geological Society, 160, 719-733

Berberian, M., 1995. Master "blind" thrust faults hidden under the Zagros folds: Active tectonics and surface morphotectonics. Tectonophysics, 241, 193-224. DOI: 10.1016/0040-1951(94)00185-C

Berberian, M., King, G.C.P. (1981) Towards a paleogeograpy and tectonic evolution of Iran. Canadian Journal of Earth Sciences, 18, 210-65. DOI: 10.1139/e81-019

Blanc, E.J.-P., Allen, M.B., Inger, S., Hassani, H. (2003) Structural styles in the Zagros Simple Folded Zone, Iran. Journal of the Geological Society, 160, 401-412. DOI: 10.1144/0016-764902-110

Bordenave, M.L. (2008) The origin of the Permo-Triassic gas accumulations in the Iranian Zagros Foldbelt and contiguous offshore areas: A review of the palaeozoic petroleum system. Journal of Petroleum Geology, 31, 3-42. DOI: 10.1111/j.1747-5457.2005.tb00087.x

Butler, R.W.H., Mazzoli, S., Corrado, S., De Donatis, M., Di Bucci, D., Gambini, R., Naso, G., Nicolai, C., Scrocca, D., Shiner, P., Zucconi, V. (2004). Applying thick-skinned tectonic models to the Apennine thrust belt of Italy--Limitations and implications. AAPG Memoir, 82, 647-667.

Cristallini, E.O., Ramos, V.A. (2000). Thick-skinned and thin-skinned thrusting in the La Ramada fold and thrust belt: crustal evolution of the High Andes of San Juan, Argentina (32 SL). Tectonophysics, 317, 205-235. DOI: 10.1016/S0040-1951(99)00276-0

Dahlstrom, C.D.A. (1969). Balanced cross sections. Canadian Journal of Earth Sciences, 6, 743-757. DOI: 10.1139/e69-069

Davis, T.L. Namson, J., Yerkes, R.F. (1989) A cross section of the Los Angeles Area: Seismically active fold and thrust belt, The 1987 W.ittier Narrows earthquake, and earthquake hazard. Journal of Geophysical Research: Solid Earth, 94, 9644-9664. DOI: 10.1029/JB094iB07p09644

English, J.M., Lunn, C.A., Ferreira, L., Yacu, G. (2015) Geologic evolution of the Iraqi Zagros, and its influence on the distribution of hydrocarbons in the Kurdistan region.

American Association of Petroleum Geologists Bulletin, 99, 231-272. DOI: 10.1306/06271413205

Falcon, N.L. (1961) Major earth-flexuring in the Zagros Mountains of south-west Iran. Quarterly Journal of the Geological Society of London, 117, 367-376. DOI: 10.1144/gsjgs.117.1.0367

Ghasemi, A., Talbot, C.J. (2006) A new tectonic scenario for the Sanandaj-Sirjan Zone (Iran). Journal of Asian Earth Sciences, 26, 683-693. DOI: 10.1016/j.jseaes.2005.01.003

Hessami, K., Koyi, H.A., Talbot, C.J., Tabasi, H., Shabanian, E. (2001) Progressive unconformities within an evolving foreland fold-thrust belt, Zagros Mountains. Journal of the Geological Society, 158, 969-982. DOI: 10.1144/0016-764901-007

Homke, S., Vergés, J., Serra-Kiel, J., Bernaola, G., Sharp, I., Garcés, M., Montero-Verdú, I., Karpuz, R., Goodarzi, M.H. (2009) Late Cretaceous-Paleocene formation of the proto-Zagros foreland basin, Lurestan Province, SW Iran. Bulletin of the Geological Society of America, 121, 7-8. DOI: 10.1130/B26035.1

Hossack, J.R. (1979). The use of balanced cross-sections in the calculation of orogenic contraction: A review. Journal of the Geological Society, 136, 705-711. DOI: 10.1144/gsjgs.136.6.0705

Jackson, J.A. (1980) Reactivation of basement faults and crustal shortening in orogenic belts. Nature, 283, 343-346. DOI: 10.1038/283343a0

Jassim, S. Z., Goff, J. C. (2006) Geology of Iraq: Dolin, Prague and Moravian Museum, Brno, Czech Republic, 341 pp.

Karim, K.H., Koyi, H., Baziany, M.M., Hessami, K. (2011) Significance of angular unconformities between Cretaceous and Tertiary strata in the northwestern segment of

the Zagros fold–thrust belt, Kurdistan. Geological Magazine, 148, 925-939. DOI: 10.1017/S0016756811000471

Lawa, F. A., Koyi, H., Ibrahim, A. (2013). Tectono-stratigraphic evolution of the NW segment of the Zagros fold-thrust belt, Kurdistan, NE Iraq. Journal of Petroleum Geology, 36, 75-96. DOI: 10.1111/jpg.12543

Kobayashi, T., Morishita, Y., Yarai, H., Fujiwara, S. (2018). InSAR-derived Crustal Deformation and Reverse Fault Motion of the 2017 Iran-Iraq Earthquake in the Northwest of the Zagros Orogenic Belt. Bulletin of the Geospatial Information Authority of Japan, in press.

Koshnaw, R. I., Horton, B. K., Stockli, D. F., Barber, D. E., Tamar-Agha, M. Y., Kendall, J. J. (2017). Neogene shortening and exhumation of the Zagros fold-thrust belt and foreland basin in the Kurdistan region of northern Iraq. Tectonophysics, 694, 332-355.

Lacombe, O., Mouthereau, F. (2002). Basement-involved shortening and deep detachment tectonics in forelands of orogens: Insights from recent collision belts (Taiwan, Western Alps, Pyrenees). Tectonics, 21. DOI: 10.1029/2001TC901018

Lacombe, O., Bellahsen, N. (2016). Thick-skinned tectonics and basement-involved fold–thrust belts: insights from selected Cenozoic orogens. Geological Magazine, 153, 763-810. DOI:10.1017/S0016756816000078

Malekzade, Z., Bellier, O., Abbassi, M. R., Shabanian, E., Authemayou, C. (2016). The effects of plate margin inhomogeneity on the deformation pattern within west-Central Zagros Fold-and-Thrust Belt. Tectonophysics, 693, 304-326. DOI: 10.1016/j.tecto.2016.01.030

Marshak, S. (1988), Kinematics of orocline and arc formation in thin-skinned orogens, Tectonics, 7, 73–86, DOI:10.1029/TC007i001p00073.

Marshak, S., Karlstrom, K., Timmons, J.M. (2000) Inversion of Proterozoic extensional faults: An explanation for the pattern of Laramide and Ancestral Rockies intracratonic deformation, United States. Geology, 28, 735-738. DOI: 10.1130/0091-7613(2000)28<735:IOPEFA>2.0.CO;2

McClay, K. R. (1989). Analogue models of inversion tectonics. Special Publications of the Geological Society of London, 44, 41-59. DOI: 10.1144/GSL.SP.1989.044.01.04

Molinaro, M., Leturmy, P., Guezou, J.C., Frizon de Lamotte, D., Eshraghi, S.A. (2005) The structure and kinematics of the southeastern Zagros fold-thrust belt, Iran: From thin-skinned to thick-skinned tectonics. Tectonics, 24, 1-19. DOI: 10.1029/2004TC001633

Mouthereau, F., Lacombe, O. (2006). Inversion of the Paleogene Chinese continental margin and thick-skinned deformation in the Western Foreland of Taiwan. Journal of Structural Geology, 28, 1977-1993. DOI:10.1016/j.jsg.2006.08.007

Mouthereau, F., Lacombe, O., Meyer, B. (2006). The Zagros folded belt (Fars, Iran): constraints from topography and critical wedge modelling. Geophysical Journal International, 165, 336-356. DOI: 10.1111/j.1365-246X.2006.02855.x

Mouthereau, F., Lacombe, O., Vergés, J. (2012) Building the Zagros collisional orogen: Timing, strain distribution and the dynamics of Arabia/Eurasia plate convergence. Tectonophysics, 532-535, 27-60. DOI: 10.1016/j.tecto.2012.01.022

Mouthereau, F., Tensi, J., Bellahsen, N., Lacombe, O., De Boisgrollier, T., Kargar, S. (2007). Tertiary sequence of deformation in a thin☐skinned/thick☐skinned collision belt: The Zagros Folded Belt (Fars, Iran). *Tectonics*, *26, TC5006.* DOI: 10.1029/2007TC002098

Navabpour, P., Barrier, E., McQuilLan, H. (2014). Oblique oceanic opening and passive margin irregularity, as inherited in the Zagros fold-and-thrust belt. Terra Nova, 26, 208-215. DOI: 10.1111/ter.12088

Nissen, E., Tatar, M., Jackson, J.A., Allen, M.B. (2011). New views on earthquake faulting in the Zagros fold-and-thrust belt of Iran. Geophysical Journal International 186, 928-944. DOI:10.1111/j.1365-246X.2011.05119.x

Obaid, A. K., Allen, M. B. (2017). Landscape maturity, fold growth sequence and structural style in the Kirkuk Embayment of the Zagros, northern Iraq. Tectonophysics, 717, 27-40. DOI: 10.1016/j.tecto.2017.07.006

Sadeghi, S., Yassaghi, A. (2016) Spatial evolution of Zagros collision zone in Kurdistan,NWIran: constraints on Arabia–Eurasia oblique convergence. Solid Earth, 7, 659-672. DOI: 10.5194/se-7-659-2016

Saura, E., Garcia-Castellanos, D., Casciello, E., Parravano, V., Urruela, A., Vergés, J. (2015) Modeling the flexural evolution of the Amiran and Mesopotamian foreland basins of NW Zagros (Iran-Iraq). Tectonics, 34, 377-395. DOI: 10.1002/2014TC003660

Sepehr, M., Cosgrove, J.W. (2004) Structural framework of the Zagros Fold-Thrust Belt, Iran Marine and Petroleum Geology, 21, 829-843. DOI: 10.1016/j.marpetgeo.2003.07.006

Shaw, J.H., Suppe, J. (1996) Earthquake hazards of active blind-thrust faults under the central Los Angeles basin, California. Journal of Geophysical Research, 101, 8623-8642. DOI: 10.1029/95JB03453

Sibson, R.H. (1985) A note on fault reactivation. Journal of Structural Geology, 7, 751-754. DOI: 10.1016/0191-8141(85)90150-6

Talbot, C.J., Alavi, M. (1996). The past of a future syntaxis across the Zagros. Geological Society of America Special Paper, 100, 89-109

Talebian, M., Jackson, J. (2002) Offset on the Main Recent Fault of the NW Iran and implications for the late Cenozoic tectonics of the Arabia-Eurasia collision zone. Geophysical Journal International, 150, 422-439. DOI: 10.1046/j.1365-246X.2002.01711.x

Talebian, M., Jackson, J.A. (2004) A reappraisal of earthquake focal mechanisms and ac- tive shortening in the Zagros mountains of Iran. Geophysical Journal International 156, 506-526. DOI: 10.1111/j.1365-246X.2004.02092.x

Tavani, S. (2012). Plate kinematics in the Cantabrian domain of the Pyrenean orogen. Solid Earth, 3, 265-292. DOI: 10.5194/se-3-265-2012

Teknik, V., Ghods, A. (2017) Depth of magnetic basement in Iran based on fractal spectral analysis of aeromagnetic data. Geophysical Journal International, 209, 1878-1891. DOI: 10.1093/gji/ggx132

Utkucu, M. (2017). Preliminary seismological report on the November 12, 2017 Northern Iran/Western Iraq earthquake. Sakarya University, DOI:10.13140/RG.2.2.17781.27364.

Vergés, J., Saura, E., Casciello, E., Fernàndez, M., Villaseñor, A., Jiménez-Munt, I., García-Castellanos, D. (2011) Crustal-scale cross-sections across the NW Zagros belt: implications for the Arabian margin reconstruction. Geological Magazine, 148, 739-761. DOI: 10.1017/S0016756811000331

Vernant, P., Nilforoushan, F., Hatzfeld, D., Abbassi, M.R., Vigny, C., Masson, F., Nankali, H., Martinod, J., Ashtiani, A., Bayer, R., Tavakoli, F., Chéry, J. (2004) Present-day crustal deformation and plate kinematics in the Middle East constrained by GPS measurements

in Iran and northern Oman. Geophysical Journal International, 157, 381-398. DOI:10.1111/gji.2004.157.issue-1.

Wang, M., Jia, D., Shaw, J.H., Hubbard, J., Lin, A., Li, Y., Shen, L. (2013) Active fault-related folding beneath an alluvial terrace in the Southern Longmen Shan range front, Sichuan basin, China: Implications for seismic hazard. Bulletin of the Seismological Society of America, 103, 2369-2385. DOI: 10.1785/0120120188

Wells, D.L., Coppersmith, K.J. (1994) New empirical relationships among magnitude, rupture length, rupture width, rupture area, and surface displacement. Bulletin of the Seismological Society of America, 84, 974-1002

Williams, G.D., Powell, C.M., Cooper, M.A. (1989) Geometry and kinematics of inversion tectonics. Special Publications of the Geological Society of London, 44, 3-15. DOI: 10.1144/GSL.SP.1989.044.01.02

Yue, L.F., Suppe, J., Hung, J.H. (2005) Structural geology of a classic thrust belt earthquake: The 1999 Chi-Chi earthquake Taiwan (Mw = 7.6). Journal of Structural Geology, 27, 2058-2083. DOI: 10.1016/j.jsg.2005.05.020

Ziegler, A.M. (2001) Late Permian to Holocene paleofacies evolution of the Arabian Plate and its hydrocarbon occurrences. GeoArabia, 6, 445-504.

---

## Short Comment (SC1) · 1 Jun 2018

Dear Stefano and co-authors. I read with interest the paper on the suggested up-dip continuation of a thrust fault from the hypocenter of the earthquake to link to the Mountain Front Flexure (MFF). The balanced section visualized nicely the anticipated structural architecture. I have some short remarks:

1) The balanced section has a local pin in Miringeh Anticline. As a consequence, you end up with some deformation SW of the Mountain Front Flexure ( i.e. 4.3km). I haven't

seen the deformation front marked on any of the maps as it is further to the SW than the MFF (cf. Verges et al, 2011 Figure 1). Is there really only 4.3 km deformation SW of the MFF? Then, wouldn't it make sense to extend the section for 5 km, have a fixed pin in the undeformed foreland and show that it restores and balances?

2) The style how the three inverting faults accommodate shortening seem all different. The style of deformation for the Marakhil and Sheykh Saleh Faults require some coupling with thin skinned decollements to distribute the shortening. The Miringeh Fault inverts straight across these potential decollement zones and then to the SW the suggested fault underneath the MFF links to this decollement at the base of the sediments again. A problem with linked thick-thin-skinned contractional systems is that the upper part of a normal fault might be decapitated by the subhorizontal movements on decollement horizons. Could that happen here, if your pin is in the foreland?

3) I find it strange that to the hinterland mainly faults invert and toward the foreland one major shortcut fault exist (the one linked to the MFF). Is that plausible? One solution could be that all major normal faults have been inverted already. Towards SW there are no more major normal faults to invert?

4) I agree, that the MFF for the Lorestan arc could well be related to basement involvement. But could you discuss alternatives and why they would not work? For other areas along the Zagros the MFF is not necessarily linked to a basement fault (see Hinsch and Bretis, 2015, Geoarabia). For the Kirkuk embayment we propose a duplex solution on multiple arguments. As a consequence we argue that the structure of the MFF is heterogeneous along-strike the Zagros. This might well be in-line with the solution presented here, given the interpreted lateral ramp at the border to the Kirkuk Embayment – but maybe it should be discussed?

I hope this comments help when reviewing your manuscript to gain higher consistency Best regards Ralph Hinsch

---

## Author Comment (AC3) · 4 Jun 2018

COMMENT The balanced section has a local pin in Miringeh Anticline. As a consequence, you end up with some deformation SW of the Mountain Front Flexure (i.e. 4.3km). I haven't seen the deformation front marked on any of the maps as it is further to the SW than the MFF (cf. Verges et al, 2011 Figure 1). Is there really only 4.3 km deformation SW of the MFF? RESPONSE The 4.3 km of shortening inferred to the SW of the MFF is highly compatible with published sections across the foreland far to the NW (e.g. Obaid and Allen, 2017). This will be mentioned in the revised version.

[Figure]

COMMENT Then, wouldn't it make sense to extend the section for 5 km, have a fixed pin in the undeformed foreland and show that it restores and balances? RESPONSE It would be fine but we have no access to subsurface data to the SE of our pin (Iraq). In addition, only Neogene sediments are exposed there, which is not particularly useful for the construction of deep cross-sections, due to the partial decoupling between Mesozoic and Cenozoic materials.

COMMENT The style how the three inverting faults accommodate shortening seem all different. The style of deformation for the Marakhil and Sheykh Saleh Faults require some coupling with thin skinned decollements to distribute the shortening. The Miringeh Fault inverts straight across these potential decollement zones and then to the SW the suggested fault underneath the MFF links to this decollement at the base of the sediments again. RESPONSE The behaviour of the Miringeh fault is well constrained by the seismic section, showing no propagation of any kind of layer-parallel shearing across it. This probably relates with the fact that this fault is in the early stage of inversion, suggesting that coupling mostly occurs due to the development of the footwall shortcut. This will be mentioned

COMMENT A problem with linked thick-thin-skinned contractional systems is that the upper part of a normal fault might be decapitated by the subhorizontal movements on decollement horizons. Could that happen here, if your pin is in the foreland? RESPONSE The observation that the major anticlines of the area sit on major basement steps (Sheykh Saleh and Marakhil anticlines), points against the activation of an important decollement level in between the Miringeh and Marakhil anticlines.

COMMENT I find it strange that to the hinterland mainly faults invert and toward the foreland one major shortcut fault exist (the one linked to the MFF). Is that plausible? One solution could be that all major normal faults have been inverted already. Towards SW there are no more major normal faults to invert? RESPONSE This is correct, in our interpretation the Miringeh fault is the innermost inherited fault and the MFF is a sort of shortcut of the inherited extensional decollement. This will be made clear.

COMMENT I agree, that the MFF for the Lorestan arc could well be related to basement involvement. But could you discuss alternatives and why they would not work? For other areas along the Zagros the MFF is not necessarily linked to a basement fault (see Hinsch and Bretis, 2015, Geoarabia). For the Kirkuk embayment we propose a duplex solution on multiple arguments. As a consequence we argue that the structure of the MFF is heterogeneous along-strike the Zagros. This might well be in-line with the solution presented here, given the interpreted lateral ramp at the border to the Kirkuk Embayment – but maybe it should be discussed? RESPONSE We will mention that the thin-skinned interpretations have been proposed for the MFF. Concerning the discussion about the heterogeneous along-strike significance of the MFF, we think that this is out of the scope of our work.